# Depression and frailty in older adults: A population-based cohort study

**Fabiana Araújo Figueiredo Da Mata**[1]*, **Marilia Miranda Forte Gomes**[2], **Jair Lício Ferreira Santos**[3], **Yeda Aparecida de Oliveira Duarte**[4], **Mauricio Gomes Pereira**[1]

1 Faculty of Medicine, University of Brasilia, Brasilia, Federal District, Brazil, 2 Faculty of Gama, University of Brasilia, Brasilia, Federal District, Brazil, 3 Department of Social Medicine, University of São Paulo, São Paulo, São Paulo, Brazil, 4 Department of Medical-Surgical Nursing, School of Nursing, University of São Paulo, São Paulo, São Paulo, Brazil

☉ These authors contributed equally to this work.
* fagfigueiredo@hotmail.com

## Abstract

### Background

Studies have shown an association between depression and frailty, even though the literature has not reached a consensus regarding how these syndromes interact. Although prospective cohort studies on this topic are still scarce, they could contribute to understanding this relationship. We aimed to observe whether depressive symptoms are risk factors for the onset of frailty among older adults living in São Paulo, Brazil.

### Methods

Prospective cohort study using the "Health, Well-being and Aging" (SABE) Study databases of 2006 and 2010. The sample was representative of the community-dwelling older adults living in São Paulo, and it is composed of non-frail men and women aged 60 years and older without cognitive decline. We calculated the frailty incidence rate between older adults with and without depressive symptoms and calculated the incidence rate ratio. Multiple analysis was carried out through Poisson regression with robust variance estimation.

### Results

The initial sample (n = 1,109) presented a mean age of 72 years (from 60 to 96) and 61.1% were women. The final sample was composed of 830 individuals, and the mean follow-up time in the study was 3.8 years. After adjusting the model, depressive symptoms did not predict the onset of frailty at follow-up.

### Conclusion

Depressive symptoms were not shown to be a risk factor for frailty among older adults living in São Paulo in this study.

**Data Availability Statement:** The data underlying the results presented in the study are available from SABE, at http://hygeia3.fsp.usp.br/sabe/.

**Funding:** This work was supported by the São Paulo Research Foundation - FAPESP, under the number 99/05125-7. The funders had no role in study design, data collection and analysis, decision to publish, or preparation of the manuscript.

**Competing interests:** The authors have declared that no competing interests exist.

## Introduction

Depression and frailty are considered common syndromes among older adults, with prevalence estimates varying from 6.5% to 25.3% for depressive symptomatology and from 4.0% to 59.1% for frailty [1–5]. Even though depression and frailty are considered two separate conditions/constructs, they share moderate overlap, meaning they can happen concomitantly in an individual [5,6].

Depression and frailty also share several commonalities (e.g., psychomotor slowness, fatigue, and weight change) [6]. Both conditions can increase the likelihood of disabilities and other causes of morbimortality in older adults [5,7]. Cross-sectional studies carried out in many geographic regions have shown some relationship between frailty and depression [8–12]. Even though similar biological mechanisms, such as subclinical cardiovascular disease and inflammation, have been hypothesized to be responsible for both frailty and depression, evidence to date shows both syndromes as different constructs [5]. However, there is still room for more investigation about the relationship and etiology of frailty and depression.

There is an agreement in the literature that some aspects of frailty are not well understood in the literature. For example, there is no consensus regarding whether physical aspects alone lead to frailty or whether psychosocial aspects also play a role in the syndrome's definition [13]. While frailty is considered a pure physical syndrome in some studies, other researchers state that psychosocial aspects are important in defining the condition [14–16].

The relationship between depressive symptoms and frailty may be bidirectional [17,18]. While there is a probability that people with frailty will develop depression (given their crescent physical limitations and lack of independence), there are cases where depressive symptoms negatively impact the body's physical functioning and may contribute to the development of frailty [1,19,20]. Although cohort studies help clarify this relationship's direction, not many studies were found to investigate this in the literature.

Therefore, this study investigates whether depressive symptoms are risk factors for the onset of frailty in older adults living in São Paulo, Brazil.

## Methods

### Study design

This study uses a prospective cohort design to analyze data collected by the "Health, Well-being and Aging" (SABE) Study, a multicentric survey that aims to investigate the life and health of older adults in seven cities from seven countries in Latin America and the Caribbean (LAC) [21]. Brazil has been collecting data since 2000, every five years on average, as part of this study in São Paulo. Our study estimates and compares the incidence of frailty in two groups of older adults with and without depressive symptoms in two points of time (2006 and 2010) as data were available. We followed the STROBE Statement checklist.

### Sampling and sample

This study included men and women aged 60 years and older without cognitive impairment and/or frailty. Data of people with cognitive impairment were excluded because of the existing evidence that cognitive impairment leads to the development of characteristics similar to the frailty syndrome [16], which could overestimate the incidence of frail older adults in this study.

The sample was divided into two groups according to the presence or absence of depressive symptoms (variable of exposure). The exposure group was composed of older adults with depressive symptoms, while the non-exposure group was composed of older adults without

depressive symptoms. Information on the sampling process originally used for the SABE Study can be found in Lebrão and Duarte, 2003 [22].

Data in this study were gathered through similar surveys conducted in 2006 and 2010, where physical and self-reported information was collected. The questionnaires used in the surveys are available on the SABE website [23].

### Study variables

**Exposure variable: Depressive symptoms.** *Depressive symptoms* were defined according to the short version of the Geriatric Depression Scale (GDS) [24,25]. The presence of depressive symptoms was considered when older adults punctuated more than five in the GDS (in such cases, they were assigned to the exposure group). People with five or fewer points on the scale were assigned to the non-exposure group.

**Outcome variable: Frailty syndrome.** We defined the *frailty syndrome* variable based on the Fried frailty phenotype, 2001 [16]. The variable was dichotomized into (a) frail and (b) non-frail older adults, where pre-frail and robust older adults were aggregated into the latter variable. To create the outcome variable, we needed to operationalize the five characteristics of the frailty phenotype. This operationalization is described next [16,26].

*1. Unintentional weight loss.* Older adults who answered that they lost at least 3 kg in the past three months received one point for frailty [27].

*2. Exhaustion.* Two questions from the *Center for Epidemiologic Studies Depression Scale* (CES-D) were used [16,28]. When older adults answered "a moderate amount of time (3 to 4 days)" or "the greater part of the time" to at least one of the two questions, they received one point for frailty. *3. Reduced walking speed.* We used a question from the SABE questionnaire that recorded the time an older adult needed to walk a 3-meter distance; the maximum amount of time considered to do this was 10 minutes. The walking speed was stratified by sex and height; the cut-off points corresponding to the lowest quintiles are shown in Table 1. People classified in the lowest quintile and those people who could not perform the test were given one point for frailty.

*4. Low level of physical activity.* We used six questions from the International Physical Activity Questionnaire (validated for older Brazilian people in 2007) [29]. The questions refer to frequency and duration of physical activities classified as low, moderate, and high intensities. The calorie loss for each activity was calculated in MET (metabolic equivalent) by multiplying each activity by the number of days per week, time (in minutes), and MET; then the calorie waste was stratified by sex and age group [26]. The calorie waste values were stratified in quintiles; people in the lowest 20% quintile received one point for frailty [16,26]. Therefore, women with calorie waste up to 327.6 Kcal/week, men with calorie waste up to 344.0 Kcal/week and not physically active older adults scored in the frailty range.

*5. Reduced handgrip strength.* Each individual could carry out the handgrip test with a dynamometer twice. The mean strength for each older person was adjusted by sex and BMI (body mass index). Afterward, the quintiles of grip strength were calculated for each BMI quartile

**Table 1. Cut-off point for the walking speed test (in seconds) by sex and height.**

| Sex | Mean height (in meters) | Cut-off point (in seconds) |
|---|---|---|
| **Male** | > 1.65 | ≥ 6 |
| | ≤ 1.65 | ≥ 6 |
| **Female** | > 1.52 | ≥ 6 |
| | ≤ 1.52 | ≥ 7 |

**Table 2. BMI quartiles adjusted by sex and their respective cut-off points for strength.**

| Male | | |
|---|---|---|
| **Quartiles** | **BMI** | **Muscle strength cut-off point** |
| 1 | 15.24–22.43 | 20.0 |
| 2 | 22.44–24.61 | 21.4 |
| 3 | 24.62–27.11 | 24.0 |
| 4 | ≥27.12 | 24.0 |
| Female | | |
| **Quartiles** | **BMI** | **Muscle strength cut-off point** |
| 1 | ≤23.14 | 13.0 |
| 2 | 23.15–26.48 | 14.0 |
| 3 | 26.49–29.76 | 14.0 |
| 4 | ≥29.77 | 14.0 |

(Table 2). Older people in the lowest (first) quintile of strength received one point for frailty [16,26]. Those who could not perform the test and who could not stand up also received one point for frailty because it was not possible to measure their heights.

The outcome variable (frailty syndrome) corresponded to the sum of the punctuation allocated to each of the five characteristics presented above. In the end, an older person could score between zero and five points for the variable frailty syndrome in this study.

Persons with punctuation equal or higher than three (they presented three or more characteristics for frailty syndrome) were classified as frail. Those people who sum to zero, one, or two points were classified as non-frail. According to Fried at al., 2001 [16], there is the pre-frailty stage (when a person presents one or two of the characteristics). However, pre-frailty was not of interest in this study as it was still considered non-frail, so older adults with one or two characteristics were considered robust in this study.

Possible confounding and adjusting variables were selected according to the literature.

**Covariables.** The selected covariables were based on self-reported answers to the SABE questionnaire in 2006. We assumed that this information was kept constant until (1) the development of frailty, (2) the date of death, or (3) the new data collection in 2010. The variables considered in this study did not present more than 1% of missing values.

The chosen variables to analyze the relationship between depressive symptoms and frailty were the most commonly encountered in the literature as potential confounders of this relationship (30). We selected ten variables of interest and grouped them into sociodemographic (sex, age, marital status, and years of schooling), health information (self-rated health, BMI, number of chronic diseases, number of medications), and lifestyle (smoking and alcohol consumption).

Older adults with cognitive decline were excluded from the sample. The operationalization of this variable was based on the Mini-Mental State Examination–MMSE (punctuation equals or is less than 12) and on the Pfeffer Scale (punctuation higher than 11) [26,30–32].

## Data analysis

The sampling process in the SABE study is complex; thus we considered this characteristic when analyzing the data. A weight was assigned to each participant according to the census sector to which they belonged. In the descriptive analysis, we compared the basal distribution of the sample characteristics between the exposure and non-exposure groups. We also used mean, standard deviation, and the chi-square test to compare the distributions between both the final sample and the lost to follow-up sample to assess the presence of bias.

In this study, the exposure time to the risk of manifesting frailty varied among the individuals; thus, the variable person-year corresponded to the exposure time in years between the date of the first interview and the manifestation or non-manifestation of frailty or occurrence of death (whichever happened first).

The incidence rate was calculated considering this follow-up time: (a) the time between the first and the second interviews for older adults who did not develop frailty, (b) half of the time between the first and the second interviews for those who developed frailty, and (c) the time between the first interview and the date of death for those who died. The incidence rate ratio was also calculated.

We conducted a univariate analysis and the variables with p-value<0.20 were selected for the multiple model [33]. Covariables known as relevant for the investigated topic were selected regardless of their statistical significance. A multicollinearity test was performed; and when correlations were more than 0.80, we chose one of the variables for not entering the multiple model [34].

A Poisson regression model with robust variance estimation using the backward regression method was chosen for the multiple analysis [35]. We considered the time of exposure to the risk of developing frailty (person-year) in the multiple model. A binomial regression test was also conducted [35].

Covariables with p-value < 0.20 in the univariate analysis entered the multiple model. However, covariables were kept in the model when they (a) presented p-value<0.05, or (b) adjusted the incidence rate ratio in at least 10%, or (c) were important for the studied outcome or, (d) improved the quality of the final model. We used the link test to check the model adjustment [36]. Calculations were performed with their respective 95% confidence intervals (95% CI) and the significance level of 5%. The software Stata® 13.0 was used for statistical analyses.

## Results

### Sample selection

This study initially included men and women aged 60 years and older (n = 1,413 in 2006). However, we excluded from this sample older adults who presented cognitive impairment and/or frailty in 2006 (n = 280) and those with missing information about these two variables (n = 14). As a result of these exclusions, in 2006 there were 1,119 non-frail older adults without cognitive decline in the sample. After excluding missing values related to depressive symptoms, the initial sample included 1,109 non-frail older adults aged 60 to 96. Fig 1 shows the flowchart representing the sample selection process in this study.

After the follow-up time,—out of the 1,109 individuals who initiated in the study– 830 survived until 2010, there were 124 who died, and 155 were lost (54 individuals were not located, 33 moved to other municipalities, 6 were institutionalized, and 62 refused to continue in the study; totalizing around 14% of lost follow-up). We considered data from 858 individuals for calculating incidence in this study (734 surviving older adults who had information about frailty at the end of 2010 and 124 non-surviving older adults).

### Sample characteristics

The initial sample (n = 1,109) presented a mean age of 72 years (from 60 to 96 years old) and 61.1% were women. Table 3 shows basal characteristics from the two groups: people with depressive symptoms and people without depressive symptoms in 2006. The distribution pattern was similar in both groups, except for the majority of older adults with depressive symptoms who self-rated their health as regular, had fewer years of education and used more medication compared with older adults without depressive symptoms.

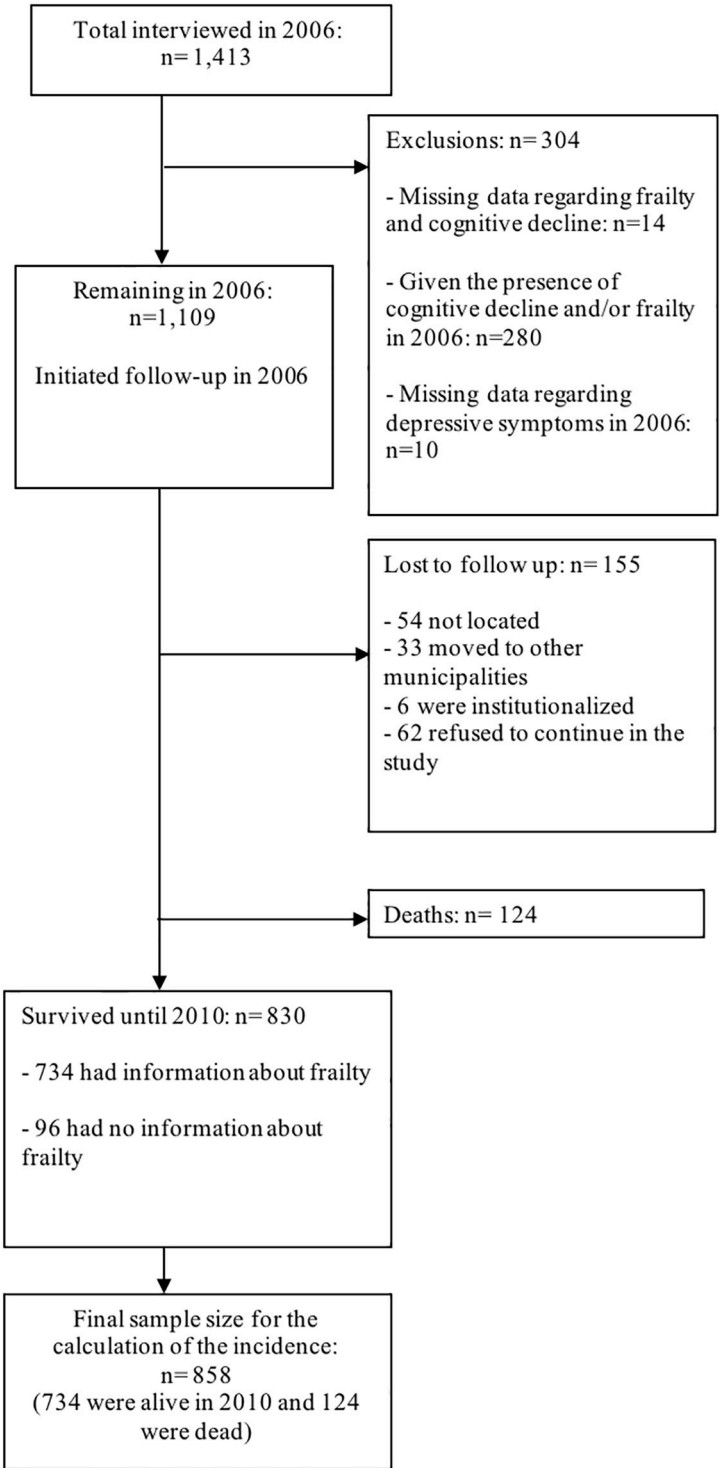

**Fig 1. Flowchart of the sample selection process.**

**Table 3. Characteristics of groups exposed and non-exposed to depressive symptoms.**

| Adjustment variables | With depressive symptoms, n = 142 | | Without depressive symptoms, n = 967 | | p-value |
|---|---|---|---|---|---|
| | Number of observations | % | Number of observations | % | |
| **A. Sociodemographic:** | | | | | |
| **Sex** | | | | | |
| Female | 107 | 75.3 | 571 | 59 | <0.001 |
| Male | 35 | 24.6 | 396 | 40.9 | |
| **Age group (years)** | | | | | |
| 60–69 | 71 | 50 | 428 | 44.3 | 0.436 |
| 70–79 | 42 | 29.6 | 323 | 33.4 | |
| 80 and older | 29 | 20.4 | 216 | 22.3 | |
| **Marital status** | | | | | |
| With partner | 70 | 49.3 | 516 | 53.4 | 0.358 |
| Without partner | 72 | 50.7 | 451 | 46.6 | |
| **Years of schooling** | | | 0 | | |
| 4 or more | 62 | 43.7 | 580 | 60 | <0.001 |
| Less than 4 | 80 | 56.3 | 387 | 40 | |
| **B. Health Information** | | | | | |
| **Self-rated health** | | | | | |
| Very good and good | 21 | 14.7 | 503 | 52 | <0.001 |
| Regular | 87 | 61.4 | 422 | 43.6 | |
| Poor and very poor | 34 | 24 | 32 | 3.3 | |
| **BMI** | | | | | |
| Normal | 63 | 44.3 | 409 | 42.3 | 0.563 |
| Underweight | 28 | 20 | 233 | 24.1 | |
| Overweight and obesity | 51 | 35.7 | 325 | 33.6 | |
| **Number of chronic diseases** | | | | | |
| None | 16 | 11.3 | 172 | 17.8 | <0.001 |
| 1 or 2 | 69 | 48.6 | 564 | 58.3 | |
| 3 or more | 57 | 40.1 | 231 | 23.9 | |
| **Number of used medication** | | | | | |
| None | 6 | 4.2 | 95 | 9.8 | <0.001 |
| 1 to 3 | 50 | 35.2 | 435 | 45 | |
| 4 or more | 86 | 60.6 | 436 | 45.1 | |
| **C. Lifestyle** | | | | | |
| **Smoking** | | | | | |
| Never smoked | 69 | 48.6 | 518 | 53.6 | <0.001 |
| Former smoker | 39 | 27.5 | 352 | 36.4 | |
| Current smoker | 34 | 24 | 97 | 10 | |
| **Alcohol consumption** | | | 0 | | |
| No | 133 | 93.7 | 913 | 94.4 | 0.717 |
| Yes | 9 | 6.3 | 54 | 5.6 | |

Note—The sample size was 1,109 in all variables, apart from the following: marital status (n = 1,108), number of medications used (n = 1,108) and BMI (n = 1,090).

Table 4 shows a comparison between the final sample (composed of individuals alive in 2010, n = 830) and the loss to follow-up sample plus non-surviving people (n = 279). In this study, the lost to follow-up sample represented a higher proportion of men, smokers and former smokers, underweight people and people older than 80. The mean follow-up time in the study was 3.8 years.

**Table 4. Comparison between the final sample (survivals in 2010) and the lost to follow-up sample.**

| Variables | Final sample in 2010, n = 830 | | Lost to follow up sample, n = 279 | | p-value |
|---|---|---|---|---|---|
| | Number of observations | % | Number of observations | % | |
| **A. Sociodemographic:** | | | | | |
| **Sex** | | | | | |
| Female | 531 | 64 | 147 | 52.7 | <0.001 |
| Male | 299 | 36 | 132 | 47.3 | |
| **Age group (years)** | | | | | |
| 60–69 | 401 | 48.3 | 98 | 35.1 | <0.001 |
| 70–79 | 280 | 33.7 | 86 | 30.8 | |
| 80 and older | 150 | 18.1 | 95 | 34 | |
| **Marital status** | | | | | |
| With partner | 445 | 53.6 | 142 | 50.9 | 0.441 |
| Without partner | 385 | 46.4 | 137 | 49.1 | |
| **Years of schooling** | | | | | |
| 4 or more | 486 | 58.5 | 156 | 55.9 | 0.44 |
| Less than 4 | 344 | 41.5 | 123 | 44.1 | |
| **B. Health information:** | | | | | |
| **Self-rated health** | | | | | |
| Very good and good | 401 | 48.3 | 123 | 44.1 | 0.27 |
| Regular | 377 | 45.4 | 132 | 47.3 | |
| Poor and very poor | 52 | 6.3 | 24 | 8.6 | |
| **BMI** | | | | | |
| Normal | 357 | 43 | 116 | 41.4 | <0.001 |
| Underweight | 170 | 20.5 | 91 | 32.7 | |
| Overweight and obesity | 304 | 36.6 | 72 | 25.8 | |
| **Number of chronic diseases** | | | | | |
| None | 141 | 17 | 47 | 16.8 | 0.465 |
| 1 or 2 | 466 | 56.1 | 167 | 59.9 | |
| 3 or more | 223 | 26.9 | 65 | 23.3 | |
| **Number of medications used** | | | | | |
| None | 79 | 9.5 | 22 | 7.9 | 0.325 |
| 1 to 3 | 353 | 42.5 | 133 | 47.5 | |
| 4 or more | 398 | 48 | 124 | 44.6 | |
| **C. Lifestyle** | | | | | |
| **Smoking** | | | | | |
| Never smoked | 461 | 55.5 | 126 | 45.2 | 0.009 |
| Former smoker | 279 | 33.6 | 112 | 40.1 | |
| Current smoker | 90 | 10.8 | 41 | 14.7 | |
| **Alcohol consumption** | | | | | |
| No | 789 | 95 | 257 | 92.1 | 0.066 |
| Yes | 41 | 4.9 | 22 | 7.9 | |

Note—The sample size was 1,109 in all variables, apart from the following: marital status (n = 1,108), number of medications used (n = 1,108) and BMI (n = 1,090).

## Incidence of frailty, univariate and multiple analyses

Among the depressive symptoms group, 12.6% of people were frail in 2010, the incidence rate (IR) was 16.4 thousand person-years (95%CI: 8.4;36.4), and the total of person-years was 444,645.5. Among older adults without depressive symptoms, the frequency of frailty in 2010

was 9.8%, the incidence rate was 12.7 thousand person-years (95%CI: 9.6;17.2) and the total of person-years was 2,997,256.5. The frailty incidence rate ratio (IRR) between individuals with and without depressive symptoms was 1.29 (95%CI: 1.26;1.32) and the total of person-years 3,441,902.1.

In the univariate analysis, five out of ten investigated variables were selected to the multiple model (Table 5): more advanced ages, living without a partner, self-rated health as regular, being overweight, and presenting three or more chronic diseases. The variables of sex and years of schooling were selected given their importance in the literature, even though they were not statistically significant. There was no multicollinearity among variables selected for the multiple model.

The incidence rate ratio in this study showed that older adults with depressive symptoms presented a 29% greater risk of developing frailty than older adults without these symptoms (IRR = 1.29; 95%CI: 1.26;1.32). Estimates from the adjusted multiple model showed the IRR = 1.21 (95%CI: 0.62;2.36), see Table 6. The link test resulted in a p-value equal to 0.772, indicating a good adjustment of the chosen model.

## Discussion

This study investigated the hypothesis that depressive symptoms are risk factors for frailty. The results showed a higher onset of frailty among older adults with depressive symptoms compared with those without these symptoms. However, after adjusting the model, depressive symptoms did not predict frailty's onset over an average of 3.8 years of follow-up.

A controversy in the literature concerns which conceptual model frailty belongs to. While some researchers affirm frailty has physical and psychological origins [14], others support the idea that frailty is purely a physical condition [16]. The results of this study seem to agree with the definition that frailty is a purely physical syndrome since, after adjusting the model, depressive symptoms did not appear to be risk factors for the onset of frailty. In addition, it is well known that depression and frailty are correlated and sometimes they are considered as the same condition [37], but as separate constructs. In this context, the findings of this study may: (a) support the hypothesis that depression and frailty are concurrent conditions in which one could be a component of the other. Even though depression has not been identified as a risk factor for frailty, they may occur concomitantly in individuals. (b) There is a need to further investigate frailty as a risk factor for depression.

Our findings agree with the results of the following investigations. The majority of the cohort studies included in a systematic review pointed out depression as a risk factor for frailty in the unadjusted analysis; however, after adjustment, the relationship between the studies disappeared [38]. Depressive symptoms did not seem to influence the incidence of frailty in a prospective cohort study conducted in the United States since there was no difference in the incidence of frailty between depressive women and negatively affected women [39]. As a consequence of the lack of consensus toward the definition of frailty, it is usual to find the terms "physical limitation"s and "functional disabilities" as synonymous with the frailty syndrome. In this sense, a panel study from the United States investigated the direction of the relationship between depressive symptoms and physical limitations over three years. The authors concluded that depressive symptoms had not been shown as risk factors for physical limitation, although the inverse relationship has been observed [40].

On the other hand, depressive symptoms have been associated with incident frailty in some investigations. A cohort study carried out with 40,659 older women in 40 clinical centers over three years in the United States showed a positive association between depressive symptoms and incident frailty [41]. Cohort studies included in a systematic review also observed that the

**Table 5. Univariate analysis among covariables and the risk of frailty.**

| Variables | Incidence ratio | Confidence Interval (95%) | p-value |
|---|---|---|---|
| A. Sociodemographic: | | | |
| **Sex** | | | |
| Male | 1 | | 0.880 |
| Female | 0.96 | (0.57–1.62) | |
| **Age group (years)** | | | |
| 60–69 | 1 | | |
| 70–79 | 2.61 | (1.47–4.62) | <0.001 |
| 80 and older | 4.94 | (2.93–8.33) | <0.001 |
| **Marital status** | | | |
| With partner | 1 | | |
| Without partner | 1.54 | (0.89–2.64) | 0.114 |
| **Years of schooling** | | | |
| 4 or more | 1 | | |
| Less than 4 | 1.35 | (0.77–2.35) | 0.277 |
| B. Health information | | | |
| **Self-rated health** | | | |
| Very good and good | 1 | | |
| Regular | 1.62 | (0.99–2.65) | 0.056 |
| Poor and very poor | 1.54 | (0.54–4.40) | 0.411 |
| **BMI** | | | |
| Normal | 1 | | |
| Underweight | 0.89 | (0.39–2.05) | 0.796 |
| Overweight and obesity | 1.52 | (0.81–2.87) | 0.186 |
| **Number of chronic diseases** | | | |
| None | 1 | | |
| 1 or 2 | 1.36 | (0.58–3.20) | 0.470 |
| 3 or more | 2.01 | (0.75–5.35) | 0.160 |
| **Number of medications used** | | | |
| None | 1 | | |
| 1 to 3 | 0.58 | (0.18–1.80) | 0.346 |
| 4 or more | 1.3 | (0.45–3.72) | 0.617 |
| C. Lifestyle | | | |
| **Smoking** | | | |
| Never smoked | 1 | | |
| Former smoker | 0.89 | (0.49–1.63) | 0.725 |
| Current smoker | 1.01 | (0.39–2.62) | 0.974 |
| **Alcohol consumption** | | | |
| No | 1 | | |
| Yes | 0.61 | (0.13–2.86) | 0.531 |

Note—The sample size was 1,109 in all variables, apart from the following: marital status (n = 1,108), number of medications used (n = 1,108) and BMI (n = 1,090).

incidence of frailty varied directly according to the severity of depression [5]. According to a prospective cohort study conducted in the United States, women with positive affect towards life presented half of the risk of manifesting frailty compared with women with negative affect [39]. Similarly, researchers observed that positive affect was associated with reducing the risk of manifesting frailty among African American women [42].

**Table 6. Frailty unadjusted and adjusted incidence rate ratio in the groups with and without depressive symptoms (n = 735).**

| Depressive symptoms | Frailty | | | | Unadjusted ratio (95%CI)* | Adjusted ratio** (95%CI)* |
|---|---|---|---|---|---|---|
| | Yes | | No | | | |
| | N | % | N | % | | |
| Yes | 12 | 1.6 | 83 | 11.3 | 1.29 (1.26;1.32) | 1.21 (0.62;2.36) |
| No | 63 | 8.6 | 577 | 78.5 | | |

* 95%CI: 95% Confidence Interval.

** Adjusted by sex, age, schooling and number of chronic diseases.

Differences in the definitions and methodological approaches adopted by researchers may influence the outcomes we found in the literature. Data from an American prospective cohort revealed that depression increased the risk of frailty when frailty was classified in a broader definition that involved physical, nutritional, cognitive, and sensorial dimensions [43].

Some studies have also investigated depressive symptoms as risk factors for conditions related to or considered to be similar to frailty, such as functional decline [44,45]. For instance, cohort data from an American study (nine years of follow-up) pointed out a high level of depressive symptoms as risk factors for moderate and severe functional decline, mainly among older women [46].

## Study limitations and strengths

Our study has several limitations. We used the variable *depressive symptoms*, which is not the same as analyzing depressive disorders and their different degrees of severity. Thus the results of this paper need to be interpreted with caution. Using the cut-off point for the variable *weight loss* as 3kg does not seem to be appropriated as 3kg may represent different body proportions according to each individual's weight. Instead of using a fixed number of kilograms, a percentage of weight loss in the past year (e.g., 5%) could be used.

In this study, the impossibility of reassessments over the follow-up time prevented us from controlling for potential interchanges between the exposure and non-exposure groups, even though this limitation is known to be a problem inherent to many cohort studies. It was also not possible to gather information about the presence of depressive symptoms in other moments of life. Therefore, the study's information relates to the health status of each individual during data collection.

Another limitation is that some of the data were self-reported, and this might have introduced bias and lesser accurate information. A possible sample selectivity bias hampered that the variable *basic activities of daily living* was considered for the chosen model in the study. Furthermore, the lack of consensus in the literature on whether frailty and difficulty in carrying out basic activities of daily living are or are not the same condition may have contributed to inconsistent results found in some statistical tests in this study (S1 and S2 Tables). Therefore, the variable basic activities of daily living was not considered in this study.

The lack of statistical significance in the final regression model may be due to the small number of cases in the study sample. The potential presence of residual confounding requires caution when interpreting the results. Another point is that the loss of follow-up was proportionally higher among people aged 80 and older compared with the initial sample. At the same time, this is the age group with the highest incidence ratio for frailty in the univariate analysis. Therefore, had this loss to follow-up not happened, there could have been room for a higher incidence rate ratio in this study. Finally, the study's external validity is limited, as our findings only refer to the population of São Paulo.

On the other hand, prospective cohort studies investigating this theme are scarce in the literature. Therefore, our findings may present more robust information from a country experiencing an accelerated ageing process that is in need of investigations. This study uses a formal definition of frailty to allow more precise comparisons of its results with the literature. It also was conducted with a representative sample of older adults living in the city of São Paulo, a city with the highest absolute number of older adults in Brazil and with the most diverse older population, as a result of the national and international immigration process [21].

In this study, depressive symptoms were not considered risk factors for the onset of frailty among older adults living in São Paulo. This result may support the theory that frailty is a physical syndrome instead of a multidimensional condition. There is room for more studies on this topic with larger and more diverse populations in Brazil and abroad.

## Supporting information

**S1 Table. Poisson multiple regression model: Initial and final models by the backward regression method.**
(DOCX)

**S2 Table. Poisson multiple regression model considering the variable basic activities of daily living: Initial and final models, by the backward regression method.**
(DOCX)

## Acknowledgments

We thank the SABE study's principal investigator and coordinator Maria Lucia Lebrao, *in memoriam*.

## Author Contributions

**Conceptualization:** Fabiana Araújo Figueiredo Da Mata, Mauricio Gomes Pereira.

**Data curation:** Jair Lício Ferreira Santos, Yeda Aparecida de Oliveira Duarte.

**Formal analysis:** Fabiana Araújo Figueiredo Da Mata, Marilia Miranda Forte Gomes, Jair Lício Ferreira Santos, Mauricio Gomes Pereira.

**Funding acquisition:** Yeda Aparecida de Oliveira Duarte.

**Methodology:** Fabiana Araújo Figueiredo Da Mata, Marilia Miranda Forte Gomes, Jair Lício Ferreira Santos, Mauricio Gomes Pereira.

**Project administration:** Fabiana Araújo Figueiredo Da Mata, Mauricio Gomes Pereira.

**Resources:** Marilia Miranda Forte Gomes, Mauricio Gomes Pereira.

**Supervision:** Marilia Miranda Forte Gomes, Mauricio Gomes Pereira.

**Visualization:** Fabiana Araújo Figueiredo Da Mata, Marilia Miranda Forte Gomes, Jair Lício Ferreira Santos, Mauricio Gomes Pereira.

**Writing – original draft:** Fabiana Araújo Figueiredo Da Mata.

**Writing – review & editing:** Fabiana Araújo Figueiredo Da Mata, Mauricio Gomes Pereira.

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
