## [Decision Letter · Decision Letter 0]

29 Dec 2020

PONE-D-20-35568

Depression and frailty in older adults: a population-based cohort study

PLOS ONE

Dear Dr. Da Mata,

Thank you for submitting your manuscript to PLOS ONE. After careful consideration, we feel that it has merit but does not fully meet PLOS ONE’s publication criteria as it currently stands. Therefore, we invite you to submit a revised version of the manuscript that addresses the points raised during the review process.

We look forward to receiving your revised manuscript.

Kind regards,

Yiqiang Zhan

Academic Editor

PLOS ONE

Journal Requirements:

Reviewers' comments:

Reviewer's Responses to Questions

**Comments to the Author**

1. Is the manuscript technically sound, and do the data support the conclusions?

Reviewer #1: Yes

Reviewer #2: Yes

2. Has the statistical analysis been performed appropriately and rigorously? 

Reviewer #1: No

Reviewer #2: Yes

3. Have the authors made all data underlying the findings in their manuscript fully available?

Reviewer #1: No

Reviewer #2: Yes

4. Is the manuscript presented in an intelligible fashion and written in standard English?

Reviewer #1: Yes

Reviewer #2: Yes

5. Review Comments to the Author

Reviewer #1: Need to specify numbers and percentage for each variables not only percentage.

-the p value in the final sample should be for each variable to identify significant confounders in logistic regression

- Alphabetical and grammatical adjustment.

Reviewer #2: It is a cohort study that was correctly delineated and conducted. The authors controlled the possible confounding variables and performed the correct statistical treatment. The limitations of the study were clearly presented.There is a small error on page 9, line 194, the value 1119 should be corrected to 1129 (1423 - 280 -14). As the only suggestion I would recommend the inclusion of a comment on the possible impact of the loss of segment of individuals aged 80 and over. The loss in this segment was proportionally higher and at the same time it is the age group with the highest incidence ratio in the univariate analysis.

6. PLOS authors have the option to publish the peer review history of their article (what does this mean?). If published, this will include your full peer review and any attached files.

Reviewer #1: No

Reviewer #2: No

---

## [Author Response · Author response to Decision Letter 0]

28 Jan 2021

Reviewer #1 

1) Need to specify numbers and percentage for each variables not only percentage. 

Reply: Thank you for your comment. We included numbers in the tables where the information was missing (Table 3 and Table 4). 

2) The p value in the final sample should be for each variable to identify significant confounders in logistic regression. 

Reply: We have added the p-values to the regression tables in the Supporting Information (S1 Table and S2 Table).

3) Alphabetical and grammatical adjustment. 

Reply: Thank you for the suggestion. The paper has been reviewed by a professional English proofreader, who has revised the writing of the paper. 

Reviewer #2 

1) There is a small error on page 9, line 194, the value 1119 should be corrected to 1129 (1423 - 280 -14).

Reply: Thank you for this comment. When revising the paper, we noticed that the error refers to a typo in the sample size, which should be 1,413 instead of 1,423. We have changed this in the line 273. 

2) As the only suggestion I would recommend the inclusion of a comment on the possible impact of the loss of segment of individuals aged 80 and over. The loss in this segment was proportionally higher and at the same time it is the age group with the highest incidence ratio in the univariate analysis.

Reply: We thank you for this suggestion. We have added a discussion of this point in the line 531.

---

## [Decision Letter · Decision Letter 1]

15 Feb 2021

Depression and frailty in older adults: a population-based cohort study

PONE-D-20-35568R1

Dear Dr. Da Mata,

We’re pleased to inform you that your manuscript has been judged scientifically suitable for publication and will be formally accepted for publication once it meets all outstanding technical requirements.

Kind regards,

Frank Y Zhan

Academic Editor

PLOS ONE

Additional Editor Comments (optional):

Reviewers' comments:

Reviewer's Responses to Questions

**Comments to the Author**

1. If the authors have adequately addressed your comments raised in a previous round of review and you feel that this manuscript is now acceptable for publication, you may indicate that here to bypass the “Comments to the Author” section, enter your conflict of interest statement in the “Confidential to Editor” section, and submit your "Accept" recommendation.

Reviewer #1: All comments have been addressed

Reviewer #2: All comments have been addressed

2. Is the manuscript technically sound, and do the data support the conclusions?

Reviewer #1: Yes

Reviewer #2: Yes

3. Has the statistical analysis been performed appropriately and rigorously? 

Reviewer #1: N/A

Reviewer #2: Yes

4. Have the authors made all data underlying the findings in their manuscript fully available?

Reviewer #1: Yes

Reviewer #2: Yes

5. Is the manuscript presented in an intelligible fashion and written in standard English?

Reviewer #1: Yes

Reviewer #2: Yes

6. Review Comments to the Author

Reviewer #1: regarding the manuscript "Depression and frailty in older adults: a population-based cohort study"PONE-D-20-35568R1 i accept for publication.

Reviewer #2: (No Response)

7. PLOS authors have the option to publish the peer review history of their article (what does this mean?). If published, this will include your full peer review and any attached files.

Reviewer #1: No

Reviewer #2: No

---

## [Editor Report · Acceptance letter]

22 Feb 2021

PONE-D-20-35568R1 

Depression and Frailty in Older Adults: A Population-Based Cohort Study 

Dear Dr. Da Mata:

I'm pleased to inform you that your manuscript has been deemed suitable for publication in PLOS ONE. Congratulations! Your manuscript is now with our production department. 

Kind regards, 

on behalf of

Dr. Frank Y Zhan 

Academic Editor

PLOS ONE